# Neighbourhood property value and type 2 diabetes mellitus in the Maastricht study: A multilevel study

David Consolazio[1,2,3]*, Annemarie Koster[1,2], Simone Sarti[4], Miranda T. Schram[5,6,7], Coen D. A. Stehouwer[5,6], Erik J. Timmermans[8], Anke Wesselius[9,10], Hans Bosma[1,2]

1 Department of Social Medicine, Maastricht University, Maastricht, The Netherlands, 2 CAPHRI Care and Public Health Research Institute, Maastricht University, Maastricht, The Netherlands, 3 Department of Sociology and Social Research, Università degli Studi di Milano-Bicocca, Milan, Italy, 4 Department of Political and Social Science, Università degli Studi di Milano, Milan, Italy, 5 Department of Internal Medicine, Maastricht University Medical Centre+, Maastricht, The Netherlands, 6 Cardiovascular Research Institute Maastricht (CARIM), Maastricht University, Maastricht, The Netherlands, 7 Heart and Vascular Center, Maastricht University Medical Center+, Maastricht, The Netherlands, 8 Department of Epidemiology and Biostatistics, Amsterdam UMC, VU University Medical Center, Amsterdam Public Health Research Institute, Amsterdam, The Netherlands, 9 Department of Complex Genetics, Maastricht University, Maastricht, The Netherlands, 10 NUTRIM School of Nutrition and Translational Research in Metabolism, Maastricht University, Maastricht, The Netherlands

* d.consolazio@maastrichtuniversity.nl

**Data Availability Statement:** Data are unsuitable for public deposition due to ethical restriction and privacy of participant data. Data are available from The Maastricht Study for any interested researcher

## Abstract

### Objective

Low individual socioeconomic status (SES) is known to be associated with a higher risk of type 2 diabetes mellitus (T2DM), but the extent to which the local context in which people live may influence T2DM rates remains unclear. This study examines whether living in a low property value neighbourhood is associated with higher rates of T2DM independently of individual SES.

### Research design and methods

Using cross-sectional data from the Maastricht Study (2010–2013) and geographical data from Statistics Netherlands, multilevel logistic regression was used to assess the association between neighbourhood property value and T2DM. Individual SES was based on education, occupation and income. Of the 2,056 participants (aged 40–75 years), 494 (24%) were diagnosed with T2DM.

### Results

Individual SES was strongly associated with T2DM, but a significant proportion of the variance in T2DM was found at the neighbourhood level (VPC = 9.2%; 95% CI = 5.0%–16%). Participants living in the poorest neighbourhoods had a 2.38 times higher odds ratio of T2DM compared to those living in the richest areas (95% CI = 1.58–3.58), independently of individual SES.

who meets the criteria for access to confidential data. The Maastricht Study Management Team (research.dms@mumc.nl) and the co-author A. Koster may be contacted to request data.

**Funding:** This study has been supported by the European Regional Development Fund as part of OP-ZUID, the province of Limburg, the department of Economic Affairs of the Netherlands (grant 310.041), Stichting Weijerhorst, the Pearl String Initiative Diabetes, the Cardiovascular Center Maastricht, Cardiovascular Research Institute Maastricht (CARIM), School for Nutrition, Toxicology and Metabolism (NUTRIM), Stichting Annadal, Health Foundation Limburg and by unrestricted grants from Janssen, Novo Nordisk and Sanofi. The regional association of General Practitioners (Zorg in Ontwikkeling (ZIO)) is gratefully acknowledged for its contribution to The Maastricht Study, enabling the invitation of individuals with T2DM by using information from its web-based electronic health record.

**Competing interests:** This study received funding from Janssen-Cilag B.V, Novo Nordisk Farma B.V. and Sanofi-Aventis Netherlands B.V. There are no patents, products in development or marketed products to declare. This does not alter the authors' adherence to all the PLOS ONE policies on sharing data and materials, as detailed online in the guide for authors. Hans Bosma serves as guest editor for the Health Inequities and Disparities Call for papers. This does not alter our adherence to PLOS ONE policies on sharing data and materials.

## Conclusions

Neighbourhood property value showed a significant association with T2DM, suggesting the usefulness of area-based programmes aimed at improving neighbourhood characteristics in order to tackle inequalities in T2DM.

## Background

With an increase in the number of people with diabetes from 108 million in 1980 to 422 million in 2014 [1], T2DM—representing about 90% of cases of diabetes—is a growing health problem worldwide. It is known that people with a SES defined by low education, occupation and income, have a higher risk of T2DM compared to people with higher SES. Behavioural risk factors, such as poor diet, low levels of physical activity, and smoking—which are known to be more common in less advantaged individuals—have been traditionally related to diabetes outcomes. Accordingly, the prevalence of T2DM shows large inequalities, disproportionally affecting deprived populations [2]. Moreover, T2DM prevalence is also related to the geographical context where people live, which can shape individuals' health outcomes through different pathways, resulting in a clear patterning of the disease at various geographical levels. A review and meta-analysis of the existing literature reports that low levels of education, occupation and income—used as SES measures—were associated with an increased risk of T2DM in high-, middle- and low-income countries [3]. Another strand of research has focused on the role played by the environment in which people live in contributing to determine diabetes outcomes. Studies in a review and meta-analysis of the literature focus on the characteristics of the built environment (green spaces, walkability, food environment, air and noise pollution), providing useful insights as regards the structural characterizations of the place of living in relation to diabetes outcomes [4]. However, they tend to neglect the role of individual socioeconomic conditions, which are often not taken into account nor adjusted for as confounders. Similarly, studies focused on the association between individual SES and diabetes outcomes rarely consider the spatial characterization of the phenomenon.

The physical and social environment in which people live influences their lifestyle, behaviours and opportunities, and ultimately also their health [5]. Processes influencing T2DM outcomes take place at different levels in individuals' lives, and each of these levels should be included in analyses aimed at estimating the effects of social circumstances and the surrounding environment.

In this study, we relied on individual self-reported measures of educational level, occupational status, and household income as indicators of individual SES, while we used the average property value of the neighbourhood as an indicator of neighbourhood SES. Rather than merely reflecting attributes of the individual real estate (e.g. size, quality, and the like), the average property value of a neighbourhood has proven to be mostly the product of area attractiveness [6] and might thus be considered a valid indicator of area SES. Accordingly, the use of property value as a proxy measure for SES has proven efficient in relation to many health outcomes, both at the individual [7,8,9] and contextual levels [10,11]. It has been suggested that measuring neighbourhood SES by using the traditional approach with aggregate measures of education, occupation, and income may not reflect the complex mechanisms interrelated in defining the characteristics of an area [12], while property value—encompassing in its evaluation also housing location and characteristics of the built environment in which homes are located—may capture the environmental and material aspects of SES in broader terms [13].

Thus, by using cross-sectional data from the Maastricht Study and matching them with aggregate-level neighbourhood data from Statistics Netherlands (CBS), we analysed neighbourhood variation in T2DM rates in the area studied and its independence from individual SES in order to evaluate the association between living in a low property value neighbourhood and T2DM.

## Research design and methods

### Study population

For our analysis, we used data from The Maastricht Study, an observational prospective population-based cohort study. The rationale and methodology have been described previously [14]. In brief, the Maastricht Study focused on the etiology, pathophysiology, complications and comorbidities of T2DM and was characterized by an extensive phenotyping approach. The study uses state-of-the-art imaging techniques and extensive biobanking to determine the health status in a population-based cohort of 10,000 individuals with T2DM individuals oversampled. Eligible for participation were all individuals aged between 40 and 75 years and living in the southern part of the Netherlands. Participants were recruited through mass media campaigns and from the municipal registries and the regional Diabetes Patient Registry via mailings. Recruitment was stratified according to known T2DM status, with an oversampling of individuals with T2DM, for reasons of efficiency.

The present study includes cross-sectional data from the first 3,451 participants in the Maastricht Study who completed the baseline survey between November 2010 and September 2013. The examinations of all participants were performed within a time window of three months. The study has been approved by the Maastricht University and the Academic Hospital's medical ethical committee (NL31329.068.10) and the Ministry of Health, Welfare and Sports of the Netherlands (Permit 131088-105234-PG). All participants gave written informed consent.

From the overall sample of 3,451 respondents, 1,395 (40.4%) were excluded due to missing data in one or more of the covariates (879 equivalent income, 77 educational level, 627 occupational status, 76 property value), or because of reporting type 1 diabetes (n = 37) or other types of diabetes (n = 4). The final sample size was 2,056 (59.6% of initial sample). We used 2-way t-test to determine whether the included group differed from the excluded one in key characteristics. The percentage of respondents with T2DM in the final sample included in the analysis was 24.0%, compared to 35.5% of the excluded population (P-value<0.001). The mean age of respondents in the included group was 58.9 years, compared to 61.0 years in the excluded one (P-value<0.001). The percentage of female respondents in the final sample was 47.2%, compared to 50.6% in the excluded group (P-value = 0.048). The mean neighbourhood property value was 222,174€ in the final sample included in the analysis, compared to 219,801€ of the excluded population (P-value = 0.666).

### Measures

**Diabetes status.** T2DM status is defined according to the 2006 WHO diagnostic criteria of glucose tolerance status [15]. All participants underwent a standardized 7-point OGTT after overnight fasting. Blood samples were collected at baseline, and 15, 30, 45, 60, 90 and 120 minutes after consumption of the 75g glucose drink. Participants who were insulin-dependent and participants with a fasting glucose level higher than 11.0 mmol/l (as determined by finger prick) did not undergo this test. Prediabetes was defined as IFG (fasting plasma glucose 6.1–6.9 mmol/l and 2-h plasma glucose <7.8 mmol/l), IGT (fasting plasma glucose <7.0 mmol/l and 2-h plasma glucose ≥7.0—< 11.1 mmol/l) or both [16]. T2DM was defined by fasting

plasma glucose ≥7.0 mmol/l or 2-h plasma glucose ≥11.1 mmol/l. Participants on diabetes medication and without type 1 diabetes were also considered as having T2DM.

**Individual SES.** The study sets out self-reported information about participants' SES, including educational level, occupational status, and income level.

*Educational level* was reported through seven ordinal categories: 1) no education, primary education not completed, primary education, 2) lower vocational education, 3) lower general education, 4) intermediate vocational education, 5) higher secondary education, 6) higher professional education, 7) university education.

*Occupational status* was reported by means of a continuous measure, according to the criteria of the International Socioeconomic Index (ISEI-08) of occupational status [17], which ranks occupational positions also by the average level of education and average earnings of job holders.

*Household income* was adjusted for household size. Household income was divided by the square root of household size, implying, for instance, that a household of four persons has twice the needs of a single-person household, as suggested by equivalent scales of the Organisation for Economic Co-operation and Development (OECD) [18].

The three individual SES measures were normalized for reasons of efficiency and to allow a comparison between the magnitude of the effect of each indicator on T2DM risk.

**Neighbourhood property value.** Information concerning socioeconomic characteristics of the neighbourhood of residence came from CBS Statistics Netherlands (*Centraal Bureau voor de Statistiek*). CBS routinely collects a wide range of indicators regarding the territory and its inhabitants, which are made available every year at three geographical levels: neighbourhoods (average area size in the study area: 1.8 km$^2$), districts (9.7 km$^2$), and municipalities (50.8 km$^2$).

We used average neighbourhood property value as proxy for neighbourhood SES. In the Netherlands, property values are assessed every year according to the criteria of the Law on the Valuation of Property (*Wet Waardering Onroerende Zaken*) [19]. The evaluation is implemented by the single municipalities, and the data are made freely available by CBS (see: www. cbsinuwbuurt.nl), at the three geographical levels. Here, we opted for neighbourhood, since property values are likely to be more homogenous at this lowest level. We divided the original continuous variable, indicating neighbourhood average property value in Euro, into quartiles, in order to have an immediately understandable comparison between people living in disadvantaged rather than advantaged neighbourhoods.

Each participant's residential address was attributed to the corresponding neighbourhood using conversion files provided by CBS [20]. The neighbourhood code of residence was then used to link the Maastricht Study data with CBS data, so that it was possible to match individual and contextual information for each respondent. For each neighbourhood, property value was assigned by computing the mean of the four years considered in the analysis (from 2010 to 2013). The correlation between the property values of the neighbourhoods in the different years of analysis was always close to one, meaning that for each neighbourhood there was almost no variation over the years and that it was reasonably possible to rely on the average value without biasing the results. The study included 82 neighbourhoods, with an average of 25 cases for each (minimum: 1; maximum: 135).

**Confounders.** Both sex and age at the year of visit were used as covariates.

## Statistical analysis

Given the hierarchical structure of the data (individuals nested in neighbourhoods), multilevel regression models [21] were used to assess simultaneously the effect of individual and

contextual characteristics, enabling estimation of the effect of the neighbourhood of residence on the outcome, independently of individual SES.

First, the spatial heterogeneity in T2DM rates was assessed using a multilevel binary logistic empty model (model 1), which enabled measurement of the extent to which the probability of having T2DM varies from one area to another. The variance partition coefficient (VPC) revealed the proportion of variability in the outcome at each level of analysis, providing a first description of the spatial distribution of the disease in the study area and evidencing the existence of a possible contextual dimension for the phenomenon studied. Second, the model was integrated with predictors at level-1 (individual SES) to investigate the extent to which area level differences were explained by the individual composition of the areas (model 2). Third, the level-2 predictor was added to check if neighbourhood property value was associated with T2DM independently of individual SES, i.e. to assess the existence of a contextual effect for T2DM in the population studied (model 3). In all the main analyses, random intercept models were fitted; multi-collinearity was checked before running the models. Additional analyses were performed with random slope models for individual SES measures and cross-level interactions. Finally, as reported more in detail in the discussion, sensitivity analyses were conducted to assess the choice of the predictors, outcomes and area units used, repeating the analysis with different operationalizations.

The multilevel logistic regression models were estimated with the Likelihood Estimation method using STATA 15 software (the analyses were also repeated with MLwiN 3.02).

## Results

Table 1 shows the distribution of the variables considered at the individual and neighbourhood levels in the models developed. Most of individuals reporting T2DM were males (73.5%) rather than females (26.5%). T2DM rates increased with age and decreased with educational level, occupational status and household income. Neighbourhoods from higher to lower property values showed proportionally higher rates of T2DM.

Table 2 shows the main results of the analyses performed. In model 1, the VPC of 9.2% informed us that, even if most of the variance was found at the individual level, there was a significant variation in T2DM outcome at the neighbourhood level. In model 2, we considered all individual predictors to account for differences in T2DM outcomes, without including any contextual variable. Among the measures of individual SES considered, both educational level (OR = 0.50; CI = 0.28–0.87) and occupational status (OR = 0.51; CI = 0.28–0.91) were inversely associated with T2DM, whilst the effect of household income on the outcome was not statistically significant. In model 3 we included all individual-level predictors, introducing the contextual variable 'property value of the neighbourhood of residence' as level-2 predictor. The results showed that, after controlling for all individual socioeconomic characteristics, people living in neighbourhoods with the lowest property values were more than twice as likely to have T2DM, as compared to people living in the better-off neighbourhoods (OR = 2.38; CI = 1.58–3.58). Significance and direction of level-1 predictors remained substantially the same on moving from model 2 to model 3. The decreasing VPC from model 1 (9.2%) to model 3 (2.2%) indicated that the predictors included in each step were able to explain most of the contextual variance, as confirmed by the decreasing AIC, indicating a better fit of the final model with all level-1 and level-2 predictors, as compared with the previous models.

We also tested models with random slope for individual SES indicators and with cross-level interactions between individual SES and neighbourhood property value. However, none of these models was statistically significant, meaning that the effect of individual SES was the

Table 1. Percentage distribution of the variables (n = 2,056).

|  | Total percentage(n = 2,056) | T2DM Percentage(n = 494) |
|---|---|---|
| **Sex** | | |
| Female | 47.2 | 26.5 |
| Male | 52.8 | 73.5 |
| **Age** | | |
| 40–53 | 26.1 | 15.6 |
| 54–59 | 23.8 | 20.5 |
| 60–65 | 26.2 | 26.5 |
| 66–75 | 24.0 | 37.5 |
| **Educational Level** | | |
| University education | 10.5 | 5.7 |
| Higher professional education | 32.2 | 25.9 |
| Intermediate vocational, Higher secondary | 29.8 | 30.4 |
| Primary, lower general/vocational | 27.6 | 38.1 |
| **Occupational status (ISEI08 score)** | | |
| 88.9–70.6 | 24.8 | 19.0 |
| 70.5–56.1 | 25.3 | 22.7 |
| 56.0–39.1 | 23.7 | 22.7 |
| 39.0–13.2 | 26.2 | 35.6 |
| **Household Income (€)** | | |
| 6,000–2,437 | 24.2 | 18.2 |
| 2,386–1,888 | 24.9 | 23.5 |
| 1,875–1,509 | 22.8 | 23.1 |
| 1,502–424 | 28.2 | 35.2 |
| **Property Value (€)** | | |
| 581,000–262,000 | 24.8 | 17.2 |
| 261,000–226,000 | 26.0 | 22.5 |
| 225,000–169,000 | 23.6 | 23.3 |
| 168,000–125,000 | 25.6 | 37.0 |

same in each neighbourhood and that the contextual effect of property value was the same for all individuals, regardless of their own SES (S1–S6 Tables).

Additionally, we assessed the impact of the choice of different indicators for the same models through a sensitivity analysis. First, running the same models with the 4-digit postcode (average area size in the study area: 5.2 km$^2$)–instead of neighbourhood (1.8 km$^2$)–as level-2 unit and with a measure of neighbourhood SES based on education, occupation and income (instead of property value) led to results similar to those reached with neighbourhood as geographical unit and property value as contextual variable (S7 Table). This suggests that the results were not influenced by the choices made in terms of area unit as well as the type of neighbourhood SES indicators. Second, analyses with the same predictors and level-2 unit but with continuous outcomes (fasting and 2-h plasma glucose tolerance status), confirmed the statistically significant association between neighbourhood property value and T2DM after controlling for age, sex and individual SES (S8 and S9 Tables). This provides evidence that our findings were not just due to the dichotomization of the outcome. Finally, running the same models including only neighbourhoods with at least 10 respondents led to analogous results (S11 Table), confirming that the results in the main models were not influenced by the inclusion of second-level units with few respondents.

**Table 2. Multilevel logistic regression of T2DM (0 = no, 1 = yes).** N = 2,056.

| | Model 1 | | Model 2 | | Model 3 | |
| | AIC: 2231.84 VPC: 9.2% | | AIC: 2030.71 VPC: 4.9% | | AIC: 2016.54 VPC: 2.2% | |
| | Odds Ratio | 95% CI | Odds Ratio | 95% CI | Odds Ratio | 95% CI |
|---|---|---|---|---|---|---|
| Intercept | 0.31 | [0.25, 0.37] | 0.06 | [0.02, 0.16] | 0.04 | [0.01, 0.11] |
| Age | | | 1.05 | [1.04, 1.07] | 1.05 | [1.04, 1.07] |
| **Sex** | | | | | | |
| Male | | | 1.00 | - | 1.00 | - |
| Female | | | 0.31 | [0.24, 0.39] | 0.31 | [0.25, 0.40] |
| **Educational Level** | | | 0.50 | [0.28, 0.87] | 0.53 | [0.30, 0.93] |
| **Occupational Status** | | | 0.51 | [0.28, 0.91] | 0.53 | [0.29, 0.95] |
| **Household Income** | | | 0.52 | [0.20, 1.32] | 0.69 | [0.27, 1.75] |
| **Property Value** | | | | | | |
| Extremely high | | | | | 1.00 | - |
| Moderately high | | | | | 1.14 | [0.76, 1.71] |
| Moderately low | | | | | 1.27 | [0.85, 1.91] |
| Extremely low | | | | | 2.38 | [1.58, 3.58] |

## Discussion

Our results showed a statistically significant association of neighbourhood property value with T2DM outcome, over and above sex, age and individual SES. The independence from SES implies that living in a poor area matters not only for individuals with fewer resources but also for the better off. Thus, people with low SES suffer the cumulative disadvantage of being exposed to the risks deriving from their personal conditions as well as to those deriving from the context in which they live.

Studies investigating the association of neighbourhood characteristics or SES with T2DM have obtained similar findings also in other contexts. A multilevel study of small-area SES in south-eastern France, reported a significantly higher prevalence of diabetes in the more deprived areas independently of individual SES [22]. An Australian study reported a lower risk of T2DM in greener neighbourhoods, even after controlling for demographic and cultural factors [23]. Similarly, a study in Canada reported that neighbourhood walkability was inversely associated with the development of diabetes [24], while a Swedish study suggested that local food environment was associated with T2DM [25]. In the United States, a series of studies reported an association of neighbourhood resources supporting physical activity and healthy diets with a lower incidence of T2DM, metabolic syndrome and insulin resistance [26,27,28,29,30]. Similarly to these studies, the results achieved here are specific of the context studied, and less likely to be applicable to lower- and middle-income counties.

Since property value has proved to predict higher rates of T2DM for people living in least valuable neighbourhoods independently of their education, occupation and income, we speculated on the possible pathways connecting the ecological level to the health outcome, following the debate between a material and a psycho-social explanation for health inequalities [31,32,33]. Least valuable neighbourhoods are possibly less equipped with supermarkets and grocery stores selling healthy food, walkable paths, proximity to green spaces and physical activity amenities, all aspects of the built environment which have been proven to be associated with higher obesity and T2DM rates [34,35,36]. In addition to this material explanation, other scholars have focused on a psychosocial one [37,38], according to which socioeconomic inequalities may be detrimental to health outcomes due to higher levels of chronic stress resulting from the psycho-social

impact of the perceived relative social position [39, 40]. In the case of T2DM, this may be related to physiological and metabolic alterations due to stress response, including overstimulation of the neuroendocrine system, which could influence the development of the disease [39]. People living in unfavourable neighbourhoods compare themselves with those living in more affluent areas, feeling inadequate and fostering processes of exclusion and stigmatization. In this sense, the average property value could be more than just a proxy for neighbourhood SES and accessibility to resources through availability from the built environment, making an intrinsic contribution itself to the social and spatial patterning of T2DM outcomes.

We also know that some people living in less advantaged areas may easily go to other neighbourhoods to grant themselves access to resources missing in their resident context. Lower SES people may be limited in their mobility, which restricts their access to health-protecting or health-enhancing resources in other neighbourhoods. Hence, we should likely be cautious in assuming that everyone might have potential access to every service and amenities in the area where he or she is located. People with less purchasing power are likely more prone to rely on services and amenities available in the proximity of their residence; the better off usually have better opportunities to go elsewhere to obtain desired facilities. Accordingly, if the issue of mobility would have to be considered, it would possibly widen the already existing socioeconomic inequalities in access to services and eventually also health.

As far as methodological choices are concerned, three issues should be mentioned. First, dealing with a contextual analysis of the social determinants of health, we are aware that a choice of different area units of analysis might have changed our results significantly. We dealt with the so-called *Modifiable Area Unit Problem* by running the same analysis with a larger area unit (4-digit postcode) and with a traditional measure of neighbourhood SES based on education, occupation, and income—thus making it possible to check also for the reliability of property value as proxy for neighbourhood SES—and the results were completely in line with those shown in the models reported (S7 Table). Data for this analysis came from the Netherlands Institute of Social Research, which provides SES scores for each 4-digit postcode area in the Netherlands for specific years [41,42].

Second, starting from two continuous measures of level of glucose in blood, in the analysis reported we opted for a model with a dichotomous outcome, without considering prediabetes as a separate category, in line with the American Diabetes Association's suggestion not to consider prediabetes as a clinical entity itself [43]. However, we also tested models considering prediabetes, separately running two binomial models comparing prediabetes with normal glucose and T2DM with normal glucose. Contrary to our expectations, the first comparison indicated that a low property value did not discriminate between the prediabetes and normal glucose categories (S10 Table). We are not sure about the reasons for the absence of an association with prediabetes, however, the linear models with continuous outcomes confirmed the presence of a territorial gradient in T2DM (S8 and S9 Tables), suggesting that the lack of relationship could be driven by the small sample size in the segmented analysis.

Third, only the comparison between living in extremely high and extremely low property value neighbourhoods was statistically significant. This is because we opted to compare the adverse effects on T2DM of living in a deprived neighbourhood instead of an affluent one. However, on choosing the lowest category as reference in our model, every comparison was statistically significant, resulting in a gradually incremental protective effect of living in a better neighbourhood (S12 Table). However, in both cases the extremely low property value quantile appeared to be substantially different from the other three, suggesting that interventions aimed at reducing spatial inequalities in T2DM onset in Southern Limburg should foremost be directed towards the improvement of the most deprived areas. The improvement to a more feasible intermediate level would already result in a substantial reduction of area-based health inequalities.

We should also highlight possible limitations of this study. First, the cross-sectional nature of the data used prevents us from assessing the direction of causation in our findings, even if it is less likely that having T2DM could affect residential choices and mobility. Reverse causality is possible, however, in the literature selective pathways at the origins of health inequalities are known to play a minor role as compared to the possibility that SES influences health outcomes, and not vice-versa [44]. Second, T2DM is the outcome of life-course processes which encompassed social and contextual exposure long before the moment in which people were involved in the study. For instance, the neighbourhood of residence in earlier life may have had a bigger effect than the current one. We had no data to keep track of this eventual residential mobility, but the area of study is reported to be one with low internal and external mobility [14], so that relocations should not represent an issue of great concern. Third, we had to exclude from the analysis cases for which information on education, occupation or income was missing. This could be a source of bias in the analysis if it altered the real distribution of the variables (S13 Table reports the distribution of the variables before excluding cases with missing data). Finally, we are aware that there are some other factors playing a fundamental role in shaping social and spatial inequalities in T2DM, which have not been included in our analysis. Other components of individual SES (e.g. health behaviours, parental background, disease familiarity, etc.) or of neighbourhood characteristics (e.g. availability of services and amenities, etc.), as well as other factors (e.g. social support, social cohesion, social networks, household composition, etc.) may intervene in the association between SES and T2DM. Nonetheless, our models were adjusted for the most pertinent indicators among those available, avoiding the risks of over-adjusting and underestimating the real effect of neighbourhood SES on T2DM onset.

Nonetheless, this study has also several strengths. The quality and the structure of the data used allowed us to measure each indicator at the proper level of analysis, without incurring the typical fallacies encountered when variables at one level are derived from data collected for units at a different level. In regard to the results, we found a very large odds ratio predicting higher T2DM rates for individuals living in neighbourhoods in the lowest property value quartile compared to those living in the better-off areas. Since we stringently controlled for individual SES, we may have faced over-adjustment [45], given that we cannot exclude a causal pathway between individual SES and neighbourhood property values, in either direction. Thus, the real association between neighbourhood SES and individual SES could be even stronger than the one reported, due to a possible underestimation. Finally, we conducted numerous sensitivity analyses, and they all confirmed our findings: the results from the model with a different area unit and another measure of neighbourhood SES provide stronger evidence for the hypotheses tested, suggesting through different operationalizations of the indicators that the results are not due to chance.

## Conclusions

This study has shown that living in neighbourhoods with low property value is associated with higher rates of T2DM, even after controlling for individual SES. Therefore, improving local contexts where people live (e.g. providing greener and more walkable spaces and healthier food environments) and mitigating inequalities between neighbourhoods could reduce T2DM rates and inequalities. Nonetheless, addressing social stratification at the individual level remains the most important strategy to pursue in order to reduce social inequalities in health [46,47]. Focusing too closely on the context may induce neglect of disadvantaged individuals living in affluent areas, whilst concentrating solely on individuals may lead to disregard of some contextual and environmental factors playing an important role in the origin of health inequalities.

## Supporting information

**S1 Table. Multilevel logistic regression of T2DM (0 = no, 1 = yes).** N = 2,056. Random slope for educational level (covariance unstructured).
(DOCX)

**S2 Table. Multilevel logistic regression of T2DM (0 = no, 1 = yes).** N = 2,056. Random slope for occupational status (covariance unstructured).
(DOCX)

**S3 Table. Multilevel logistic regression of T2DM (0 = no, 1 = yes).** N = 2,056. Random slope for household income (covariance unstructured).
(DOCX)

**S4 Table. Multilevel logistic regression of T2DM (0 = no, 1 = yes).** N = 2,056. Cross-level interaction between educational level and property value (covariance unstructured).
(DOCX)

**S5 Table. Multilevel logistic regression of T2DM (0 = no, 1 = yes).** N = 2,056. Cross-level interaction between occupational status and property value (covariance unstructured).
(DOCX)

**S6 Table. Multilevel logistic regression of T2DM (0 = no, 1 = yes).** N = 2,056. Cross-level interaction between household income and property value (covariance unstructured).
(DOCX)

**S7 Table. Multilevel logistic regression of T2DM (0 = no, 1 = yes) with 4-digits postal code and neighbourhood SES.** N = 2,056.
(DOCX)

**S8 Table. Multilevel linear regression of fasting plasma glucose tolerance status.** N = 2,077.
(DOCX)

**S9 Table. Multilevel linear regression of 2-h plasma glucose tolerance status.** N = 1,942.
(DOCX)

**S10 Table. Multilevel logistic regression of comparing a) normal glucose levels individual with prediabetes (N = 1,562); b) normal glucose levels individual with T2DM (N = 1,739); c) prediabetes with T2DM (N = 811).**
(DOCX)

**S11 Table. Multilevel logistic regression of T2DM (0 = no, 1 = yes).** N = 1,908. Only neighbourhoods with at least 10 respondents included*.
(DOCX)

**S12 Table. Multilevel logistic regression of T2DM (0 = no, 1 = yes).** N = 2,056. Property value reverse coded.
(DOCX)

**S13 Table. Percentage distribution of the variables before excluding cases with missing data.**
(DOCX)

## Author Contributions

**Conceptualization:** Annemarie Koster, Hans Bosma.

**Data curation:** Erik J. Timmermans.

**Formal analysis:** David Consolazio, Simone Sarti, Hans Bosma.

**Investigation:** David Consolazio.

**Methodology:** David Consolazio, Simone Sarti, Hans Bosma.

**Project administration:** Annemarie Koster, Hans Bosma.

**Supervision:** Annemarie Koster, Simone Sarti, Miranda T. Schram, Coen D. A. Stehouwer, Erik J. Timmermans, Anke Wesselius, Hans Bosma.

**Writing – original draft:** David Consolazio.

**Writing – review & editing:** David Consolazio.

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
