## [Decision Letter · Decision Letter 0]

27 Feb 2020

PONE-D-19-33108

Neighbourhood Property Value and Type 2 Diabetes Mellitus in the Maastricht Study: a Multilevel Study

PLOS ONE

Dear Mr Consolazio,

Thank you for submitting your manuscript to PLOS ONE. After careful consideration, we feel that it has merit but does not fully meet PLOS ONE’s publication criteria as it currently stands. Therefore, we invite you to submit a revised version of the manuscript that addresses the points raised during the review process.

We would appreciate receiving your revised manuscript by Apr 11 2020 11:59PM. To enhance the reproducibility of your results, we recommend that if applicable you deposit your laboratory protocols in protocols.io, where a protocol can be assigned its own identifier (DOI) such that it can be cited independently in the future. For instructions see: http://journals.plos.org/plosone/s/submission-guidelines#loc-laboratory-protocols

We look forward to receiving your revised manuscript.

Kind regards,

Eyal Oren, Ph.D.

Academic Editor

PLOS ONE

Journal Requirements:

2. Please correct your reference to "p=0.000" to "p<0.001" or as similarly appropriate, as p values cannot equal zero.

"This study has been supported by the European Regional Development Fund as part of OP-ZUID, the province of Limburg, the department of Economic Affairs of the Netherlands (grant 31O.041), Stichting Weijerhorst, the Pearl String Initiative Diabetes, the Cardiovascular Center Maastricht, Cardiovascular Research Institute Maastricht (CARIM), School for Nutrition, Toxicology and Metabolism (NUTRIM), Stichting Annadal, Health Foundation Limburg and by unrestricted grants from Janssen, Novo Nordisk and Sanofi. The regional association of General Practitioners (Zorg in Ontwikkeling (ZIO)) is gratefully acknowledged for its contribution to The Maastricht Study, enabling the invitation of individuals with T2DM by using information from its web-based electronic health record."

We note that you received funding from a commercial source: Janssen, Novo Nordisk and Sanofi

6. We note that Hans Bosma serves as guest Editor for the Health Inequities and Disparities Call for papers, could you please provide an amended Competing Interest Statement to declare this. Please also confirm that this does not alter your adherence to all PLOS ONE policies on sharing data and materials, by including the following statement: "This does not alter our adherence to PLOS ONE policies on sharing data and materials.” (as detailed online in our guide for authors http://journals.plos.org/plosone/s/competing-interests). If there are restrictions on sharing of data and/or materials, please state these.

Reviewers' comments:

Reviewer's Responses to Questions

**Comments to the Author**

1. Is the manuscript technically sound, and do the data support the conclusions?

Reviewer #1: Partly

Reviewer #2: Partly

2. Has the statistical analysis been performed appropriately and rigorously? 

Reviewer #1: Yes

Reviewer #2: Yes

3. Have the authors made all data underlying the findings in their manuscript fully available?

Reviewer #1: No

Reviewer #2: Yes

4. Is the manuscript presented in an intelligible fashion and written in standard English?

Reviewer #1: Yes

Reviewer #2: Yes

5. Review Comments to the Author

Reviewer #1: This paper essentially builds on the previously published analyses of the cross-sectional relationship between socioeconomic status and diabetes prevalence in the baseline data collection for the Maastrict Cohort Study. Another recent publication (included as reference 2) explores in much more detail the association with a wide range of individual level markers of socioeconomic status. This analysis uses the potential to link the dataset to neighbourhood property values to extend the use of this dataset to explore the association with this area-level variable.

Whilst the approach to analysis and presentation of findings seems appropriate and clear, both the rationale for choosing property value as a proxy of environmental determinants of diabetes risk and the caveats in terms of assuming this is a causal association in the interpretation of the findings could be more adequately addressed as suggested below:

1. As property value reflects both an element related to the size and quality of individual properties and an element related to the perceived attractiveness of the area (eg local amenities, access to public transport, quality of local green spaces etc) it is difficult without any information on, or adjustment for, the former to know the extent to which property value reflect area rather than property specific characteristics. This makes it difficult to be confident of the authors assertion that property value is measuring area SES rather being a proxy for individual variable not adjusted for.

2. It appears from the previously published analysis that there may be other confounding variables and residual confounding by individual factors (for which property value may be a proxy) cannot be excluded as explanation for findings.

3. The lack of a relationship with prediabetes is also a finding that suggests it is less likely that this cross-sectional association is due to areas characteristics associated with property values being on the causal pathway to diabetes. Authors need to at least discuss the potential explanation of reverse causality with health status (including obesity and diabetes) being on a causal pathway that results in an individual living in an area with lower property values.

Reviewer #2: Overall comment

An interesting study with ‘daring’ interpretation that perhaps require further data and explanation.

Background

Further information and examples of the different environmental and material aspects of the different categories of property in particular between that of extreme high and extreme low would aid understanding of this variable on T2DM. Further comparisons to other countries and discussion has been quite well done under the Discussion section but probably requiring a more attention to the differences between the two extreme categories, as shown by the result.

Lines 67-70: the statement is less accurate as the review reported that most of the included studies that adjusted for lifestyle behaviours and psychosocial factors had insignificant effects except one study solely in women.

Methods

1. Slightly more information on “extensive phenotyping approach” may help understanding of the Maastricht Study.

2. Disclose also the differences between the included and excluded respondents from the aspects of NPV based on the CBS Statistics Netherlands. If this information was lost (n=76), please explain how did this happen because isn’t it a readily available data?

3. Were all patients in the regional Diabetes Patient Registry selected and invited? How were the oversampling of people with T2DM executed and completed?

4. How good is the regional Diabetes Patient Registry in terms of coverage of the people with a diagnosis of T2DM in the Southern part of the Netherlands? Could you provide a comparison between those included respondents and those in the registry at large from the aspects of NPV (if this is possible)? Supplemental Table 7 shows that the extreme groups of the occupational status and property value quartiles could be over- and under-represented.

5. I believe the municipal registries hold the total residential information. Could you provide a comparison between those included respondents and those in the regions at large from the aspects of NPV (if this is possible)?

6. Line 158: please provide more detail on the normalization procedure.

Results

Suggest to put the total (n=…) on Table 1 caption itself and for the T2DM column.

Providing key reason/s for the 3 sensitivity analyses would be appreciated here although are quite well discussed in the Discussion. Is the third one for a support of (small) sample size?

Discussion

Could NPV (and its personal SES indicators) a correlate to health behaviours that has an effect on T2DM? Isn’t it established that people of higher SES more available and capable for healthier lifestyles? Greener and more walkable spaces and healthier food environments will not benefit anyone who is not motivated to self-regulation and self-care, or unavailable and not affordable for them. The former (NPV) may act as the external motivator and the later (personal SES) may complement as the internal motivator and both would function through regular/consistent health behaviours on T2DM. The association could be bidirectional and inherited through generations (shared personal and social traits) in the majority. What would be a more plausible explanation for the association between NPV and T2DM?

Lines 311-316: do discuss a bit more on this statistical observation. The extremely low quartile property value seems to be substantially different from all the others (Supplemental Table 6). From another perspective, concluding and suggesting to achieve the external features of the moderately low property value is probably more achievable by many stakeholders to also enjoy its health benefits similar to that in the extremely high property value.

It is probably good to remind the readers that the results are likely not/less applicable to lower/middle income countries.

Are ‘greener and more walkable spaces and healthier food environments’ true observed differences between the different NPV in the southern Netherlands? Are not these relatively accessible to almost all occupants?

6. PLOS authors have the option to publish the peer review history of their article (what does this mean?). If published, this will include your full peer review and any attached files.

Reviewer #1: No

Reviewer #2: Yes: Boon-How Chew

---

## [Author Response · Author response to Decision Letter 0]

8 May 2020

All the issues raised by the editor and the reviewers have been addressed in the rebuttal letter attached.

---

## [Decision Letter · Decision Letter 1]

26 May 2020

Neighbourhood Property Value and Type 2 Diabetes Mellitus in the Maastricht Study: a Multilevel Study

PONE-D-19-33108R1

Dear Dr. Consolazio,

We are pleased to inform you that your manuscript has been judged scientifically suitable for publication and will be formally accepted for publication once it complies with all outstanding technical requirements.

With kind regards,

Eyal Oren, Ph.D.

Academic Editor

PLOS ONE

Additional Editor Comments (optional):

Reviewers' comments:

Reviewer's Responses to Questions

**Comments to the Author**

1. If the authors have adequately addressed your comments raised in a previous round of review and you feel that this manuscript is now acceptable for publication, you may indicate that here to bypass the “Comments to the Author” section, enter your conflict of interest statement in the “Confidential to Editor” section, and submit your "Accept" recommendation.

Reviewer #1: All comments have been addressed

Reviewer #2: All comments have been addressed

2. Is the manuscript technically sound, and do the data support the conclusions?

Reviewer #1: Yes

Reviewer #2: Yes

3. Has the statistical analysis been performed appropriately and rigorously? 

Reviewer #1: N/A

Reviewer #2: Yes

4. Have the authors made all data underlying the findings in their manuscript fully available?

Reviewer #1: Yes

Reviewer #2: Yes

5. Is the manuscript presented in an intelligible fashion and written in standard English?

Reviewer #1: Yes

Reviewer #2: Yes

6. Review Comments to the Author

Reviewer #1: Thank you for addressing reviewers' comments

Reviewer #2: (No Response)

7. PLOS authors have the option to publish the peer review history of their article (what does this mean?). If published, this will include your full peer review and any attached files.

Reviewer #1: No

Reviewer #2: Yes: Boon-How Chew

---

## [Editor Report · Acceptance letter]

29 May 2020

PONE-D-19-33108R1 

Neighbourhood Property Value and Type 2 Diabetes Mellitus in the Maastricht Study: a Multilevel Study 

Dear Dr. Consolazio:

I am pleased to inform you that your manuscript has been deemed suitable for publication in PLOS ONE. Congratulations! Your manuscript is now with our production department. 

With kind regards,

on behalf of

Dr. Eyal Oren 

Academic Editor

PLOS ONE